# Clinical and Functional Results of Cementless Unicompartmental Knee Arthroplasty with a Minimum Follow Up of 5 Years—A Consecutive Cohort of 201 Patients

**DOI:** 10.3390/jcm12041694

**Published:** 2023-02-20

**Authors:** Benjamin Panzram, Frederik Barbian, Tobias Reiner, Mustafa Hariri, Tobias Renkawitz, Tilman Walker

**Affiliations:** Department of Orthopaedics, Heidelberg University Hospital, Schlierbacher Landstrasse 200a, 69118 Heidelberg, Germany

**Keywords:** cementless unicompartmental knee arthroplasty, Oxford knee, OUKR, UKR

## Abstract

The number of unicompartmental knee replacements (UKR) is increasing. Alongside various advantages, the revision rate of cemented UKR is higher compared to total knee arthroplasty (TKR). In contrast, cementless fixation shows reduced revision rates, compared to the cemented UKR. However, most of the recent literature is based on designer-dependent studies. In this retrospective, single-center cohort study, we investigated patients who underwent cementless Oxford UKR (OUKR) between 2012 and 2016 in our hospital with a minimum follow-up of five years. Clinical outcome was evaluated using the OKS, AKSS-O, AKSS-F, FFbH-OA, UCLA, SF-36, EQ-5D-3L, FJS, ROM, pain, and satisfaction measures. Survival analysis was performed with reoperation and revision as endpoints. We included 201 patients (216 knees) for clinical evaluation. All outcome parameters increased significantly from pre- to postoperative stages. The five-year survival rate was 96.1% for revision surgery and 94.9% for reoperation. The main reasons for revision were the progression of osteoarthritis, inlay dislocation, and tibial overstuffing. Two iatrogenic tibial fractures appeared. Cementless OUKR shows excellent clinical outcome and high survival rates after five years. The tibial plateau fracture in cementless UKR represents a serious complication and requires modification of the surgical technique.

## 1. Introduction

Osteoarthritis (OA) is a degenerative joint disease with loss of the protective cartilage. With a prevalence of 16% worldwide [1] and a global increase of over 100% since 1990, it is one of the most important reasons for joint disability worldwide [1,2,3]. Although any joint can be affected, the knee is by far the most common localization of OA, occurring in over half of OA patients [4]. In about 50% of cases, only one knee compartment is affected, with the medial compartment being five to ten times more frequently affected than the lateral [5,6]. In the late stages of the disease, endoprosthetic joint replacement is a reliable treatment option. According to The German Arthroplasty Registry 2020, 124.677 primary knee arthroplasties were performed nationwide in 2019 [7]; 13.5% of these were unicompartmental knee replacements (UKR) [7]. UKR is a viable option for patients with isolated osteoarthritis in the medial or lateral compartment of the knee. UKR appears to offer some advantages over total knee replacement (TKR) such as better cost efficiency, higher return to activity rate, physiological movement, and shorter operating time as well as a reduced risk of overall complications [8,9,10,11,12,13,14,15,16,17,18]. In addition, the clinical outcome of UKR seems to achieve excellent results more frequently [19,20,21]. Although UKR implantations have increased in recent years, the numbers are still rather small compared to the potential indications for UKR, which account for up to 30–50% of cases with OA of the knee [11,22].

Despite the good clinical results, UKR shows higher revision rates compared to TKR, especially for low surgeon caseload [18,20,21,23,24,25]. However, the risk of revision seems to have decreased due to the increasing use of UKR in general and the implementation of cementless fixation [26]. One of the most commonly used UKR is the Oxford knee Phase III (OUKR) [25]. To enhance biological fixation, the cementless version uses porous titanium and a hydroxyapatite coating. In the literature, cementless fixation seems to lead to comparably good clinical outcome and survival rates while reducing revision rates [27,28,29,30,31]. Mohammad et al. presented the results of 1000 patients with a survival rate of 97.5% after a 10-year follow-up [32].

However, most of the studies in the recent literature are derived from the designing centers. Therefore, after presenting the results of approximately 200 cementless OUKR with a 2-year follow-up, the aim of this study was to further investigate the medium-term results with a minimum follow-up of 5 years [33].

## 2. Materials and Methods

### 2.1. Design

In this retrospective single-center cohort study, we identified 276 patients (304 knees) who underwent cementless OUKR at our institution between October 2012 and November 2016. All surgeries were performed by senior surgeons with profound experience in the surgical procedure. All patients were included according to the criteria described in previous publications [34,35]. The minimum follow-up was 5 years. This study was conducted in accordance with the guidelines of the 2013 revised declaration of Helsinki. All included patients gave their written consent to participate in the study. The study was conducted with the approval of the internal ethics committee (S-346/2020).

### 2.2. Clinical Outcome

To measure clinical outcome, we used the Oxford Knee Score (OKS, 0–48 points), the American Knee Society Score including the Objective (AKSS-O, 0–100 points) and Functional (AKSS-F, 0–100 points) parts, the Hannover Functional Ability Questionnaire for Osteoarthritis (FFbH-OA, 0–100%), and the University of California at Los Angeles activity Score (UCLA, 0–10). The Short Form 36 (SF-36), the European Quality of Life 5 Dimensions 3 Level Version (EQ-5D-3L, index: 0–1), and the Forgotten Joint Score (FJS, 0–100) were used to assess quality of life. Values of these scores were only recorded at the last follow-up and compared to standardized reference groups [36,37,38,39,40]. Pain was assessed on a visual analogue scale (VAS; 0 = no pain − 10 = extreme pain) as well as satisfaction (1 = “extremely satisfactory” − 5 = “unsatisfactory). We also asked whether patients would choose this treatment again (“surely yes”, “probably yes”, “probably no”, “surely not”, “unsure”). If patients were unable to take part in a follow-up examination, the data were collected by mail and telephone.

### 2.3. Survival

A Kaplan–Meier survival analysis was performed with two different endpoints: revision and reoperation. Revision was defined as any surgical procedure on the knee joint in which at least one component of the implant was replaced. Reoperation was defined as any surgical procedure of the knee joint without replacement of prosthetic components. Procedures that did not involve the knee were not counted as reoperation.

### 2.4. Statistics

IBM SPSS Statistics version 28.0.0.0 (IBM Corp., Armonk, NY, USA), Microsoft Excel version 16 (Microsoft Corp., Redmond, WA, USA) and GraphPad Prism version 9.4.1.681 (GraphPad Software, LCC, San Diego, CA, USA) were used for all statistical analysis and graphic presentation. The Wilcoxon-test was used to compare preoperative with postoperative clinical outcome values (OKS, AKSS-F, AKSS-O, FFbH-OA, UCLA, ROM, pain). The Mann–Whitney-U test was used to investigate correlation with gender. Possible correlation of clinical outcome parameters and BMI was tested with the Spearman correlation coefficient (r < |0.1| = no correlation; r between |0.1| and |0.2| = low correlation; r between |0.2| and |0.4| = moderate correlation; r between |0.4| and |1| = high correlation). To counteract the problem of multiple comparisons, we decided to set the significance level at 0.01.

## 3. Results

### 3.1. Patient Cohort

Out of the original collective of 276 consecutive patients (304 knees), 239 patients (86.60%; 261 knees) could be included. Overall, 201 patients (72.83%; 216 knees) were included in the clinical evaluation with a minimum follow-up of 60 months. Only for survival analysis, 38 patients (13.77%, 45 knees) were available due to revision surgery, death, or request not to participate in the clinical follow-up. A total of 37 patients (13.4%; 43 knees) were excluded from the study. Patients who were lost to follow-up could not be reached by phone, mail, or email. The main reason for drop-out was the wish not to further participate in the study. In two cases, patients were excluded after a fall from a tree with consecutive tibial fracture after 4 months and 16 months with a fracture of the tibia and fibula. Furthermore, one patient (1 knee) was excluded because of a postoperatively diagnosed tenosynovial giant cell tumor (TGCT) that resulted in revision to TKR after 18 months. The detailed patient collective is depicted in Figure 1.

Of the 201 patients we were able to include in the clinical follow-up, 15 patients (6.9%) received a bilateral treatment with a cementless OUKR. The mean follow-up in this cohort was 79.85 months (SD: 11.32; range: 60–105 months). At the time of surgery, the mean age was 61.29 years (SD: 9.62; range: 36–80) and at the last follow-up, it was 67.97 years (SD: 9.60; range: 43–86). The gender ratio was 1.3:1 (male/female) with 113 male (56.2%; 122 knees) and 88 female (43.8%; 94 knees) patients. 98 left (45.4%) and 118 right knees (54.6%) were included. Based on the BMI (mean: 30.572, SD: 5.465, range: 20.20–49.59), there were at least 157 patients (78.11%) classified as being overweight (BMI ≥ 25) and 106 patients (52.74%) classified as having t class 1 obesity (BMI ≥ 30).

### 3.2. Revision Surgery and Reoperation

Revision surgery was performed in 10 knees (3.83%) and two cases (0.76%) were due to inlay dislocation. The inlay was exchanged after 3 and 11 months. In the first case, the patient stepped out of the car and in the second case the leg was moved while lying down. Seven knees (2.68%) were excluded from further clinical evaluation after revision surgery. In five cases (1.92%), the OUKR was revised to TKR due to progression of OA after 11, 12, 19, 27, and 45 months. In the other two patients (0.76%), the tibial components were exchanged to a cemented version with a simultaneous exchange of the inlay due to tibial overstuffing resulting in pain and an impaired range of motion. One knee was already revised after 9 days and the second one after 16 months postoperatively. Two cases (0.76%) of tibial plateau fractures occurred. They appeared in the medial compartment without trauma after 26 days and two months after surgery. In the first case, a reoperation with only ORIF and plate fixation, and in the second case, a revision with additional exchange of the inlay were performed. Furthermore, two more reoperations were necessary: one joint manipulation under anesthesia at 3 months due to arthrofibrosis and one wound debridement after 2 weeks due to a wound infection. Figure 1 gives an overview of the cohort.

### 3.3. Survival

The cumulative five-year survival for revision was 96.1% (95%-CI: 92.8–97.9%) and for reoperation 94.9% (95%-CI: 91.4–97.0%). The Kaplan–Meier curve for implant revision is shown in Figure 2.

### 3.4. Functional Scores

The evaluation of OKS, AKSS-O, AKSS-F, FFbH-OA, UCLA, ROM, pain, and FJS is presented in Table 1. There was a highly significant improvement from the preoperative stage to the last follow-up for all values (*p* < 0.001).

**Table 1 jcm-12-01694-t001:** Functional Scores.

Score	Spalte2	PreOp	Last Follow-Up	Δ	*p*-Value
OKS	Mean	31.75	40.74	8.97	
	SD	7.34	7.49	11.88	<0.001
	IQR	26–37	37–46	2–18	
AKSS-O	Mean	50.83	82.83	31.90	
	SD	12.74	16.53	18.86	<0.001
	IQR	41–59	75–95	21–46	
AKSS-F	Mean	61.09	83.73	22.46	
	SD	20.26	20.99	24.17	<0.001
	IQR	50–80	75–100	10–40	
FFbH-OA	Mean	65.50	82.42	16.87	
	SD	17.05	19.02	19.29	<0.001
	IQR	53–78	71.25–97	6–28	
UCLA	Mean	3.28	6.33	3.14	
	SD	1.84	1.77	2.37	<0.001
	IQR	2–4	6–7	2–5	
ROM	Mean	121.06	127.38	6.77	
	SD	14.22	12.71	14.44	<0.001
	IQR	111–130	120–135	0–15	
Pain	Mean	7.17	1.99	−5.2	
	SD	1.90	2.51	2.92	<0.001
	IQR	6–9	0–3	−7–(−3)	
FJS	Mean	-	68.24	-	
	SD	-	30.15	-	
	IQR	-	45.8–95.8	-	

Mean results of the SF-36 were 72.84 (SD: 22.54) for physical function (PhyFun), 61.07 (SD: 43.65) for role limitations (physical) (RolPhy), 65.76 (SD: 30.34) for body pain (BodPai), 59.85 (SD: 20.61) for general health (GenHea), 58.08 (SD: 21.29) for Vitality (Vit), 78.54 (SD: 27.68) for social functioning (SocFun), 72.14 (SD: 41.74) for role limitations (emotional) (RolEmo) and 71.86 (SD: 20.70) for mental health (MenHea). Results of SF-36 dimensions are presented in Figure 3.

Regarding the EQ-5D-3L-dimensions, no problems were reported from 68 patients for pain/discomfort (33.8%), 149 for anxiety/depression (74.1%), 133 for mobility (66.2%), 184 for self-care (91.5%) and 149 (74.1%) for usual activities. Relative results are visualized in Figure 4 in comparison with norm values for the European population (65–74 years) [36]. The mean index score for this collective was 0.777 (SD: 0.235; median: 0.885; IQR: 0.545–0.885). The index value for reference group was 0.862 (SD: 0.202).

### 3.5. Satisfaction

Overall, 87% of the patients were at least “satisfied” with the clinical outcome, 22.7% described the result even as “extremely satisfactory”, 43.5% as “very satisfactory”, 7.4% as “satisfactory”, 7.4% as “sufficient”, and 5,6% as unsatisfactory”. A total of 93.9% of the patients would choose this treatment again.

### 3.6. BMI

We could detect a significant correlation regarding the BMI for some clinical outcome measures. OKS, AKSS-F, FFbH-OA, ROM, and UCLA as well as the dimensions PF, RP, BP, and VT of the SF-36 showed a moderate but no significant negative correlation in the last follow-up. The FJS showed a significant low negative correlation concerning the BMI.

## 4. Discussion

In this independent retrospective study, we evaluated the clinical outcome of 201 patients (216 knees) who underwent cementless medial OUKR at our institution between October 2012 and November 2016 with a minimum follow-up of 5 years.

In this study, the cumulative five-year revision-free survival rate was 96.1%, which is consistent with the results of the current literature. The New Zealand Joint Registry reports a five-year survival rate of 94.8% [25]. Manara et al. recently reported a survival rate of 94.5% after a mean follow-up of 7.9 years [41]. Nandra et al. and Hefny et al. reported a 97.8% and a 97.4% five-year survival rate [42,43]. The main reasons for revision match the results in the recent literature and registry data, with progression of OA and inlay dislocation being among the most common [25,32,43,44].

Three tibial fractures occurred in the postoperative course in our study. In two cases (0.76%), fractures appeared without a specific trauma within the first 2 months after surgery. In both patients, the implant could be preserved and ORIF was performed. One knee showed signs of an extended sagittal saw cut in the postoperative radiograph which has been considered as a risk factor for a tibial plateau fracture in a cadaveric study by Seeger and colleagues. They describe an increased incidence of fractures for cementless fixation, especially in combination with poor bone quality, due to a decreased loading capacity of the tibia with cementless fixation and an increased impaction force needed to seat the coated implant to obtain sufficient press fit [45]. Campi and Mohammed showed in their biomechanical studies that widening the keel slot during tibial preparation using a bone pic or a different saw blade may reduce the impaction force while maintaining a sufficient press fit [46,47]. After experiencing the first plateau fractures, we adjusted our surgical technique by using a wider cemented bone pick for keel preparation with no further fractures to date. However, Keppler et al. argue in a recent clinical study that usage of the bone pick itself seems to be a risk factor for subsequent tibial plateau fractures and thus should be avoided. They suggest using a mono reciprocating saw blade [48].

However, when comparing the incidence of postoperative periprosthetic tibial plateau fractures, which occur mostly within the first three months after implantation, the systematic literature review by Burger et al. did not show a significant difference between cementless (1.24%) and cemented fixation (1.58%) [49].

Regarding postoperative function, all available outcome parameters improved significantly from pre- to postoperative stages (OKS, AKSS-O, AKSS-F, FFbH-OA, UCLA, ROM, and pain). OKS improved by 8.97, AKSS-O by 31.90, AKSS-F by 22.46, and FFbH-OA by 16.87 points. ROM increased by 6.77 degrees and pain decreased by 5.2 points. After cementless OUKR mean UCLA changed from 3 (limited housework, occasionally walk) to around 6 (unlimited housework, activities such as swimming, light physical work) and shows a significant improvement regarding activity in daily life. OKS, AKSS-O and AKSS-F are well established and frequently used patient-reported outcome measures. In the literature, the five-year results for OKS, AKSS-O, and AKSS-F differ at 38.3–43, 81–94, and 75.2–83.6, respectively [32,41,42,43,50].

Regarding health-related quality of life in cementless UKR, Martin et al. reported an EQ-5D-5L index score of 0.81 after five years [27] which is similar to our study (0.77) and slightly lower than the value of the normative data (0.862) reported previously by Janssen et al. [36]. However, in every dimension, more than 90% of the patients described no problems or only some problems.

Values of the SF-36 dimensions after cementless OUKR were comparable with the German reference group for the general population between 61–70 years [37].

Concerning joint awareness after cementless partial knee replacement, our group showed an FJS of 68.2 which is lower than previously reported after UKR but still underlines the findings of the recent literature that UKR patients seem to forget their artificial joint than patients after TKR [39,40,51]. Zuiderbaan et al. reported an FJS of 59.8 two years after TKR [51].

The satisfaction rate was high; 87% of the patients were at least satisfied with the clinical outcome. Kahlenberg et al. reported in a systematic review a satisfaction rate of 80% after TKR [52].

A weakness of this study is its retrospective design, as well as a relatively low number of participants who were available for the evaluation of clinical results (201 patients, 72.83%). This may be caused by the acute pandemic situation which led many patients to avoid hospitals especially when not having problems with the implant. All data collection at the last follow-up was performed during the acute phase of the COVID-19 pandemic. Maugeri et al. reported significantly lower physical activity and mental health during the COVID-19 pandemic [53]. These circumstances may have led to poorer outcomes in some cases and should be considered when comparing with other literature. Another limitation of this study is that there was no direct control group. The comparison was made regarding current literature findings. Furthermore, we did not collect preoperative data on the SF-36 and EQ-5D-3L. A strength of this study is the rather large cohort of patients evaluated in an independent center using a wide selection of clinical outcome parameters.

## 5. Conclusions

In summary, the results of this independent retrospective cohort study show an excellent mid-term clinical outcome, as well as a high satisfaction and survival rate of the cementless OUKR. Tibial plateau fracture in cementless fixation represents a rare but severe complication. In this regard, the importance of accurate saw cuts and keel preparation should be emphasized.

## Figures and Tables

**Figure 1 jcm-12-01694-f001:**
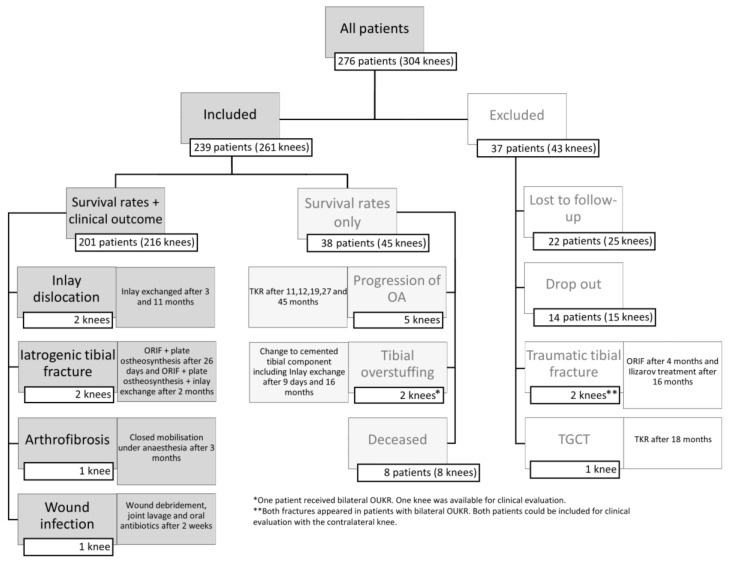
Overview—patient cohort.

**Figure 2 jcm-12-01694-f002:**
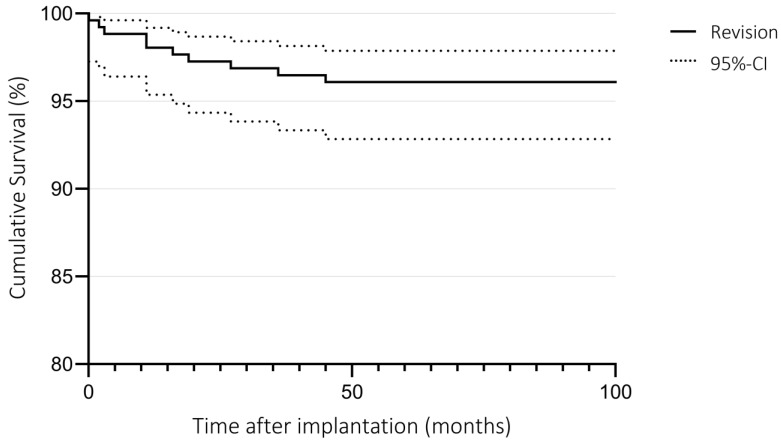
Implant survival.

**Figure 3 jcm-12-01694-f003:**
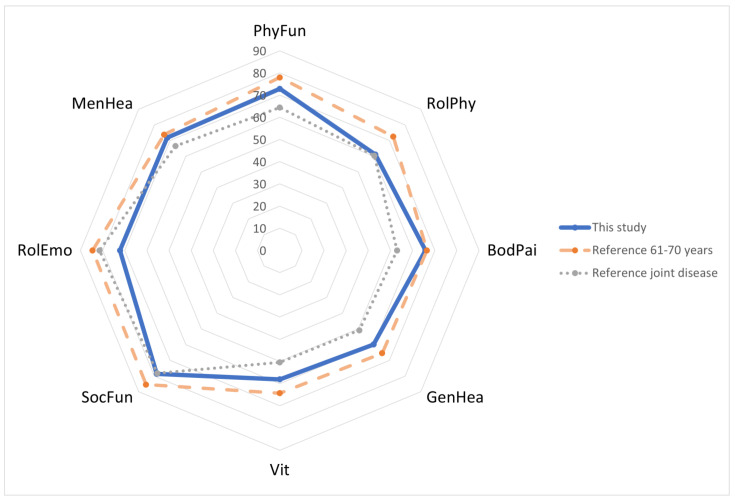
SF-36.

**Figure 4 jcm-12-01694-f004:**
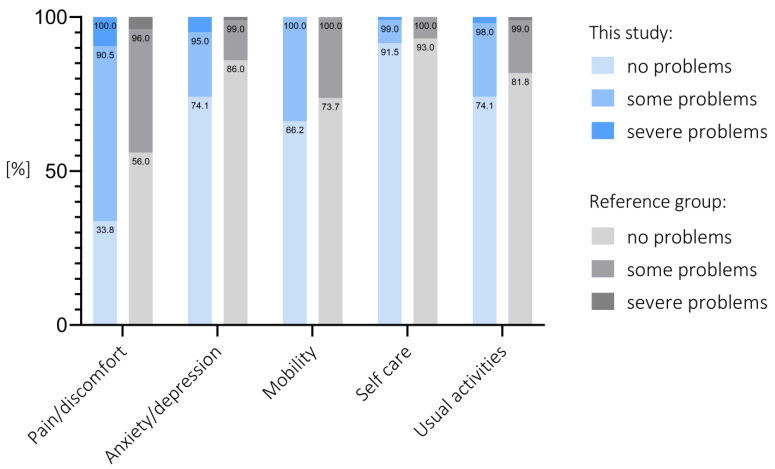
EQ-5D-3L.

## Data Availability

Data will be provided by the corresponding author upon request.

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
