# Peer review of "Clinical and Functional Results of Cementless Unicompartmental Knee Arthroplasty with a Minimum Follow Up of 5 Years—A Consecutive Cohort of 201 Patients"

_jcm, 2023, doi:10.3390/jcm12041694_

Round 1
Reviewer 1 Report
This research reports a medical report of the surgery using the cementless Oxford UKR applied on 276 patients in a medical center with a follow-up of five years. The results show good clinical outcome with a satisfaction and survival rate of the employed cementless (OUKR). The following notation should be unified: Oxford UKR and OUKR. Moreover, the English spelling needed to be approved.
Reviewer 2 Report
- minimum 5-yrs fup is in 216 knees (201 pts).... then title and abstract should be corrected
-again, patients cohort paragraph is unclear and also causes for revision surgery and/or failure have to be clarified better
Reviewer 3 Report
Dear Authors,
The manuscript is well written and interesting for other researchers, as well as surgeons. I have only few comments.
Introduction: "According to The German Arthroplasty Registry 2021, 111,365 primary knee arthroplasties were performed nationwide in 2020 [7]." - data from 2020 are probably the latest, but they can be not reliable, as the number of surgeries was probably (like in other contries) affected by COVID-19 pandemic. Instead of 2020, it would be better to cite data from 2019.
Material and Methods:
- please, divide this section into subsections to make it easier to read,
- you used non-parametric tests to evaluate a group of 300 knees - please provide the justification for this; did you checked data distribution before?
Results:
- please, number the subsections to make it easier to read,
- quality of the figures is relatively low - probably it is caused by some technical issues, but please provide figures in good quality for publication purposes.
Round 2
Reviewer 2 Report
I THINK THAT NOW THE PAPER SHOULD BE CONSIDERED FOR PUBLICATION